# Rényi Regularised Reinforcement Learning

## Abstract

Entropy regularisation has proven effective in reinforcement learning (RL) for encouraging exploration. Recent work demonstrating the equivalence between entropy regularised RL and approximate probabilistic inference suggests the potential for improving existing methods by generalising the inference procedure. We develop the Rényi regularised RL framework by using Rényi variational inference to learn a stochastic policy. We present theoretical results for policy evaluation and improvement within this new framework. Additionally, we propose two novel algorithms, $\alpha$-SAC and $\alpha$-SQL, for large-scale RL tasks. We show that these algorithms attain higher returns on games from the Atari suite relative to an entropy-regularised benchmark, SAC-Discrete.

## 1 Introduction

Success in deep reinforcement learning requires that an algorithm continually refine its behavioural policy for interacting with the environment. Such refinement often involves collapsing down the space of viable actions, as the relative values of different actions becomes clearer. However, an algorithm which interacts with the environment according to a fully deterministic policy will eventually fail to learn any further, since it will not gather additional data about alternative actions which may be superior to those currently being pursued. This is the basic problem of *exploration* in reinforcement learning. Recently, a suite of algorithms (Haarnoja et al., 2017; 2019; Christodoulou, 2019) have attempted to address this exploration problem using entropy regularisation (or more generally, KL-regularisation) which penalises policies for having very low entropy. This encourages policies to take a variety of actions, thus exploring new potential strategies in the environment.

Theoretical work has shown that the entropy regularised RL objective is equivalent to a approximate probabilistic inference problem (Levine, 2018). This insight allows us to move freely between a *regularisation view* of the entropy regularised RL problem and a corresponding *inference view* (See Fig. 1). Previous research generalising entropy regularised RL has focused on generalising the regularisation view (Yang et al., 2019). In this paper, we take a fundamentally different approach, by instead starting from the inference view and considering generalisations of the approximate inference procedure. Specifically, we will learn a policy via $\alpha$-Rényi variational inference (Li & Turner, 2016). Doing so gives rise to a novel RL objective. We prove theoretical results for this objective, and then leverage these results in the design of two novel deep RL algorithms for discrete, deterministic environments: $\alpha$-Soft Actor-Critic ($\alpha$-SAC) and $\alpha$-Soft Q-Learning ($\alpha$-SQL)[1]. We compare these with their entropy-regularised counterpart, SAC-Discrete (Christodoulou, 2019), and show that they are able to achieve a higher return on a range of Atari environments.

Fig. 1 shows the structure of the first half of the paper. In Sec. 2, we recapitulate existing results elaborating on the connection between KL-regularised RL and probablistic inference. In Sec. 3 we generalise to the Rényi regularised setting, and state core theoretical results for this setting. In Sec. 4 we use these results to design deep RL algorithms for the non-tabular setting. Finally, we compare these algorithms against an existing baseline in Sec. 5.

---

[1]A link to a GitHub repository containing implementations of these algorithms will be added for the camera ready version.

*Inference view*                                   *Regularisation view*

$$\mathbb{E}_{\tau \sim q_\pi} \left[ \beta\, G(\tau) \right] - D_{\mathrm{KL}} \left( q_\pi(\tau) \,\|\, p(\tau) \right) \longleftarrow \mathbb{E}_{\tau \sim \pi} \left[ \sum_{t=0}^{T-1} \beta\, r(a_t, s_t) - D_{\mathrm{KL}} \left( \pi(a_t \,|\, s_t) \,\|\, \pi_b(a_t \,|\, s_t) \right) \right]$$

$$\downarrow \textbf{Rényi variational inference}$$

$$\mathbb{E}_{\tau \sim q_\pi} \left[ \beta\, G(\tau) \right] - D_\alpha \left( q_\pi(\tau) \,\|\, p(\tau) \right) \longrightarrow \frac{1}{1-\alpha} \log \mathbb{E}_{\tau \sim \pi} \left[ \prod_{t=0}^{T-1} \left( \frac{e^{\beta\, r(s_t, a_t)}\, \pi_b(a_t \,|\, s_t)}{\pi(a_t \,|\, s_t)} \right)^{1-\alpha} \right]$$

Figure 1: **Relationship between KL-regularised reinforcement learning and Rényi regularised reinforcement learning.** In Sec. 2.1 we review the KL-regularised RL objective (top right). In Sec. 2.2 we move to the inference view (top left), which treats the RL problem as a structured variational inference problem. In Sec. 2.3 we explain how to generalise this inference objective to the $\alpha$-Renyi variational inference objective (bottom left). Finally, we expand this objective in terms of a policy in Sec. 3 which allows us to formulate the $\alpha$-Rényi reinforcement learning objective (bottom right).

## CONTRIBUTIONS

Our contributions are as follows:

- The introduction of a novel RL objective, the Rényi regularised RL objective.
- Theoretical results for both policy evaluation and policy improvement for the Rényi regularised RL objective.
- The introduction of two new algorithms for the Rényi regularised RL problem, $\alpha$-SAC and $\alpha$-SQL.
- Empirical evaluations of $\alpha$-SAC and $\alpha$-SQL on deterministic discrete RL tasks.

## 2 PRELIMINARIES AND RELATED WORK

### 2.1 THE KL-REGULARISED SETTING

The *entropy-regularised reinforcement learning problem* has been the subject of much attention in recent years, yielding novel state-of-the-art algorithms such as Soft Actor-Critic (SAC) (Haarnoja et al., 2019) and Soft Q-Learning (SQL) (Haarnoja et al., 2017). In short, the entropy-regularised RL problem modifies the standard RL objective by administering additional rewards to the agent for taking actions that have a low likelihood under its current policy. Below we focus on the more general *KL-regularised reinforcement learning problem*, in which penalties are administered to the agent for pursuing a policy which deviates from a *base policy* $\pi_b(a|s)$; the entropy-regularised problem can be obtained from the KL-regularised one by setting the base policy $\pi_b(a|s)$ to be the (unnormalised) uniform distribution, $\pi_b(a|s) = 1$.

We consider an episodic Markov Decision Process (MDP) described by an environment with a collection of *states* $\mathcal{S}$. The agent begins in state $s_0 \sim p_0(s_0)$, and samples an *action* $a_0$ from its policy, $a_0 \sim \pi(a|s_0)$. Following this action, the agent receives a reward $r_1 = r(s_0, a_0)$, and transitions into a new state $s_1$ according to the *dynamics* $p(s_1|s_0, a_0)$. This process then repeats, yielding a sequence of states, actions, and rewards $s_0, a_0, r_1, s_1, a_1, r_2, \ldots$. This sequence terminates at time $T$ when the agent transitions into a terminal state $s_T$. The sequence of states, actions, and rewards from the initial state to the terminate state are referred to as a *trajectory*, $\tau = (s_0, a_0, \ldots, s_T)$. The *return* of a trajectory is given by $G(\tau) = \sum_{t=1}^{T} r_t$. We refer the reader to Sutton & Barto (2020) for a more comprehensive introduction to MDPs.

In the undiscounted case, the KL-regularised reinforcement learning problem is to find a policy $\pi$ which maximises:

$$J(\pi) := \mathbb{E}_{\tau \sim \pi} \left[ \sum_{t=0}^{T-1} \beta r(a_t, s_t) - D_{\mathrm{KL}} \left( \pi(a_t|s_t) || \pi_b(a_t|s_t) \right) \right] \tag{1}$$

where $D_{\mathrm{KL}}$ is the KL-divergence, defined by:

$$D_{\mathrm{KL}}(\pi(a|s)||\pi_b(a|s)) := \int \pi(a|s) \log \left( \frac{\pi(a|s)}{\pi_b(a|s)} \right) da. \tag{2}$$

Here $\beta$ is an *inverse temperature parameter*, which dictates the trade-off between maximising the return $G(\tau)$ and minimising the KL-divergence. We call Eq. (1) the *regularisation view* of KL-regularised RL, since the KL-divergence adds a penalty term to the objective which is independent of environmental rewards. For a more extensive introduction to the KL-regularised RL problem, we refer the reader to Levine (2018).

Existing work (Yang et al., 2019) has attempted to generalise the KL-regularised RL setting by starting with the regularisation view, and substituting the KL-divergence penalty in Eq. (1) with the expectation of a more general function $\Omega$ of the policy density, yielding the objective:

$$J_\Omega(\pi) := \mathbb{E}_{\tau \sim \pi} \left[ \sum_{t=0}^{T-1} \beta r(a_t, s_t) - \Omega \left( \pi(a_t|s_t) \right) \right]. \tag{3}$$

In this paper, we adopt a fundamentally different approach. In Sec. 2.2, we show how to move to an alternative view of KL-regularised RL which we term the inference view. Under the inference view, KL-regularised RL is understood as a form of *approximate inference*. This allows us to develop novel algorithms by generalising the inference procedure, which we do in Sec. 2.3.

## 2.2 KL-REGULARISED RL AS STRUCTURED VARIATIONAL INFERENCE

*Variational Inference* (Ganguly et al., 2023) is a popular method in ML for learning approximations to the posterior of a distribution conditional on observed data. We consider a pair of variables $x$ and $z$, described by joint distribution $p(x, z)$. Conditional on an observation of $x$, we wish to learn an approximation to the posterior $p(z|x)$. In variational inference, we attempt to learn an approximate posterior $q(z) \approx p(z|x)$ by maximising the *Evidence Lower Bound (ELBo)*, given by:

$$\mathcal{L}(q) := \mathbb{E}_{z \sim q} \left[ \log \left( \frac{p(x, z)}{q(z)} \right) \right] \tag{4}$$

The ELBo can be re-expressed in the following form:

$$\mathcal{L}(q) = \log p(x) - D_{\mathrm{KL}}(q(z)||p(z|x)) \tag{5}$$

Since the KL-divergence term is non-negative, and is equal to zero precisely when $q(z) = p(z|x)$, we see that the ELBo $\mathcal{L}$ is maximised precisely when $q(z) = p(x, z)$. Accordingly, the optimisation problem of maximising the ELBo corresponds to the inference problem of computing the posterior for $z$, conditional on $x$.

The ELBo can also be written in an equivalent form – used, *e.g.*, in the Variational Auto-Encoder (Kingma & Welling, 2019) – is given by:

$$\mathcal{L}(q) = \mathbb{E}_{z \sim q} \left[ \log p(x|z) \right] - D_{\mathrm{KL}} \left( q(z)||p(z) \right). \tag{6}$$

In this form, the first term acts as an objective which encourages the approximate posterior $q(z)$ to sample latent variables $z$ under which the data $x$ has a high likelihood, while the second term can be seen as a regularisation which encourages the approximate posterior $q(z)$ to stay close to the true latent variable prior $p(z)$. It is this form we will focus on in later derivations.

Having introduced both the KL-regularised RL problem (Sec. 2.1) and variational inference, we now show how the KL-regularised reinforcement learning objective (Eq. (1)) can be understood as a special case of *variational inference*. We term this the *inference view* of KL-regularised RL. For a more complete exposition, we refer the reader to Levine (2018).

We begin by introducing an optimality variable, $\mathcal{O} \in \{0, 1\}$, whose conditional probability satisfies:

$$p(\mathcal{O} = 1|\tau) \propto \exp \left( \beta G(\tau) \right). \tag{7}$$

As before $\beta$ is an *inverse temperature parameter*, and $G(\tau)$ is the return of the trajectory $\tau$. The optimality variable is named so because it has a higher probability of being 'on', i.e. 1, the higher the

return of the trajectory. In the inference view of the KL-regularised RL problem, this variable takes the role of observed variables, $x$, while the RL trajectory, $\tau$, plays the role of the hidden variables $z$. The prior $p(\tau)$ over the trajectories is assumed to correspond to that for an RL agent implementing the base policy policy

$$p(\tau) := p_0(s_0) \prod_{t=0}^{T-1} p(s_{t+1}|s_t, a_t)\pi_b(a_t|s_t). \tag{8}$$

We will now consider using a special form of variational inference to compute an approximate posterior $q$ over trajectories, given the observation $\mathcal{O} = 1$. In *structured variational inference* (Hoffman & Blei, 2014), an assumption is made regarding the decomposition of $q(\tau)$ across the various sub-variables of $\tau$.

We derive a form for $q$ by assuming that the density over trajectories can be modified only by pursuing a different (Markovian) policy, and not by changing the environmental dynamics. The resulting expression is:

$$q_\pi(\tau) = p_0(s_0) \prod_{t=0}^{T-1} p(s_{t+1}|s_t, a_t)\pi(a_t|s_t). \tag{9}$$

In effect, Eq. (9) isolates action selection as the part of the system which is under our control, while leaving environmental dynamics untouched. Using the likelihood in Eq. (7), the prior in Eq. (8), and the approximate posterior in Eq. (9) in the form of the ELBo (Eq. (6)) yields:

$$\mathcal{L}(q_\pi) = \mathbb{E}_{\tau \sim q_\pi}[\beta G(\tau)] - D_{\mathrm{KL}}(q_\pi(\tau)||p(\tau)). \tag{10}$$

We term Eq. (10) the inference view of KL-regularised RL. Note that the KL-divergence can be simplified as follows:

$$D_{\mathrm{KL}}(q_\pi(\tau)||p(\tau)) = \mathbb{E}_{\tau \sim q_\pi}\left[\log\left(\frac{q(\tau)}{p(\tau)}\right)\right] = \mathbb{E}_{\tau \sim q_\pi}\left[\log\left(\prod_{t=0}^{T-1} \frac{\pi(a_t|s_t)}{\pi_b(a_t|s_t)}\right)\right]$$

$$= \mathbb{E}_{\tau \sim q_\pi}\left[\sum_{t=0}^{T-1} \log\left(\frac{\pi(a_t|s_t)}{\pi_b(a_t|s_t)}\right)\right] = \mathbb{E}_{\tau \sim q_\pi}\left[\sum_{t=0}^{T-1} D_{\mathrm{KL}}(\pi(a_t|s_t)||\pi_b(a_t|s_t))\right]$$

Substituting this expression into Eq. (10) demonstrates the equivalence between the inference view, Eq. (10), and the regularisation view, Eq. (1), as required. Having now introduced the inference view of KL-regularised RL, we consider a generalisation of the inference procedure and the corresponding generalisation of the KL-regularised RL problem.

## 2.3 Rényi divergence variational inference

$\alpha$-Rényi divergences (van Erven & Harremoës, 2014) form a one-parameter family of discrepancy measures between pairs of probability distributions. For $\alpha \neq -\infty, 1, +\infty$, the $\alpha$-Rényi divergence from density $q$ to density $p$ is given by:

$$D_\alpha(q||p) := \frac{1}{\alpha - 1} \log \mathbb{E}_q\left[\left(\frac{q(z)}{p(z)}\right)^{\alpha-1}\right] \tag{11}$$

This definition is extended by continuity to $-\infty, 1, +\infty$. In particular, the 1-Rényi divergence is exactly the KL-divergence, Eq. (2).

Before continuing, we briefly note some properties of the $\alpha$-Rényi divergences. Firstly, the $\alpha$-Rényi divergence is non-negative for $\alpha > 0$ and non-positive for $\alpha < 0$. For this reason we will restrict our attention for the rest of this paper to the case $\alpha > 0$. Secondly, the $\alpha$-Rényi divergence is continuous and non-decreasing as a function of $\alpha$; hence larger $\alpha$-values lead to greater penalisation. Lastly, for all $\alpha \geq 1$, $D_\alpha(q||p)$ is zero-forcing in $q$, meaning that, if $D_\alpha(q||p) < \infty$, $q = 0$ whenever $p = 0$. However, for $0 < \alpha < 1$, $D_\alpha(q||p)$ is not zero-forcing in $q$. We refer the reader to (van Erven & Harremoës, 2014) for a more in-depth discussion of Rényi divergences and their properties.

Rényi divergence variational inference (Li & Turner, 2016) generalises classical variational inference by replacing the KL-divergence appearing in the ELBo, Eq. (5), with the $\alpha$-Rényi divergence:

$$\mathcal{L}_\alpha(q) := \log(p(x)) - D_\alpha(q(z)||p(z|x)). \tag{12}$$

This is referred to as the *Variational Rényi lower bound*. Eq. (12) can be rearranged into the following equivalent form (*cf.* with Eq. (4)):

$$\mathcal{L}_\alpha(q) := \frac{1}{1-\alpha} \log \mathbb{E}_{z \sim q(z)} \left[ \left( \frac{p(x,z)}{q(z)} \right)^{1-\alpha} \right] \tag{13}$$

## 3 THE RÉNYI REGULARISED REINFORCEMENT LEARNING PROBLEM

In Sec. 2.2 we introduced the inference view of KL-regularised RL. In this section, we generalise the inference procedure by replacing the ELBo in Eq. (10) with the Variational Rényi lower bound. This yields a new family of RL objectives, parameterised by $\alpha$:

$$\mathcal{L}_\alpha(q_\pi) = \frac{1}{1-\alpha} \log \mathbb{E}_{\tau \sim q_\pi(\tau)} \left[ \left( \frac{e^{\beta G(\tau)} p(\tau)}{q_\pi(\tau)} \right)^{1-\alpha} \right]. \tag{14}$$

As in Sec. 2.2, we will once again make the *structure assumption*, Eq. (9). This gives the following objective in terms of only a policy, $\pi(a|s)$:

$$\mathcal{L}_\alpha(q_\pi) = \frac{1}{1-\alpha} \log \mathbb{E}_{\tau \sim q_\pi(\tau)} \left[ \prod_{t=0}^{T-1} \left( \frac{e^{\beta r(s_t,a_t)} \pi_b(a_t|s_t)}{\pi(a_t|s_t)} \right)^{1-\alpha} \right]. \tag{15}$$

We will refer to Eq. (15) as the *$\alpha$-Rényi reinforcement learning objective*.

Having established the $\alpha$-Rényi RL objective, we seek to develop practical deep RL algorithms for maximising this objective. To do so, we start by defining the (undiscounted) $\alpha$-soft state-value function via:

$$V_\alpha^\pi(s) := \frac{1}{1-\alpha} \log \mathbb{E}_{\tau \sim q_\pi(\tau)} \left[ \prod_{t=0}^{T-1} \left( \frac{e^{\beta r(s_t,a_t)} \pi_b(a_t|s_t)}{\pi(a_t|s_t)} \right)^{1-\alpha} \middle| s_0 = s \right], \tag{16}$$

We can learn state-value functions by leveraging Bellman recursion relationships. In App. A.1 we show that the (undiscounted) $\alpha$-soft state-value function satisfies the Bellman recursion relationship:

$$V_\alpha^\pi(s) = \frac{\beta^{-1}}{1-\alpha} \log \mathbb{E}_{a \sim \pi(a|s)} \left[ \left( \frac{e^{\beta r(s,a)} \pi_b(a|s)}{\pi(a|s)} \right)^{1-\alpha} \mathbb{E}_{s' \sim p(s'|s,a)} \left[ e^{\beta(1-\alpha) V_\alpha^\pi(s')} \right] \right]. \tag{17}$$

The convergence of practical algorithms which have their basis in recursion relationships like Eq. (17) typically rely on the introduction of a discount factor $\gamma \in (0,1)$, which reduces the value of upcoming states. Accordingly, we will introduce discounting[2] by defining the corresponding *$\alpha$-soft Bellman operator*, $\mathcal{B}_\alpha^\pi$, which acts on state-value functions via:

$$\left[\mathcal{B}_\alpha^\pi V\right](s) = \frac{\beta^{-1}}{1-\alpha} \log \mathbb{E}_{a \sim \pi(a|s)} \left[ \left( \frac{e^{\beta r(s,a)} \pi_b(a|s)}{\pi(a|s)} \right)^{1-\alpha} \mathbb{E}_{s' \sim p(s'|s,a)} \left[ e^{\beta(1-\alpha) V(s')} \right]^\gamma \right] \tag{18}$$

We also define the action of $\mathcal{B}_\alpha^\pi$ on action-value functions $Q(s,a)$ via:

$$\left[\mathcal{B}_\alpha^\pi Q\right](s,a) := r(s,a) + \gamma \frac{\beta^{-1}}{1-\alpha} \log \mathbb{E}_{a',s' \sim \pi(a'|s')p(s'|s,a)} \left[ \left( \frac{e^{\beta Q(s',a')} \pi_b(a'|s')}{\pi(a'|s')} \right)^{1-\alpha} \right] \tag{19}$$

In the limit as $\alpha \to 1$, this recovers the typical soft Bellman operators for the KL-regularised setting. We now present the first theoretical result of the paper. This result will be used to define the discounted $\alpha$-soft state- and action-value functions, and allow us to perform iterative policy evaluation to find those functions:

---

[2]The role of the discount factor in our algorithm can be understood as a regulariser to allow convergence of iterative policy evaluation protocols. See Amit et al. (2020) for further discussion.

**Theorem 1** ($\alpha$-soft policy evaluation). *Consider a finite MDP, i.e., $|\mathcal{S} \times \mathcal{A}| < \infty$. Then (for both state-value and action-value functions), for any $\gamma \in (0, 1)$, the $\alpha$-soft Bellman operator is a contraction mapping in the $\ell^\infty$ norm with contraction modulus $\gamma$. Accordingly, there exist unique fixed points, which we call the $\alpha$-soft state- and action-value functions, denoted by $V_\alpha^\pi(s)$ and $Q_\alpha^\pi(s, a)$ respectively, to which any sequence of iterates converges in the $\ell^\infty$ norm. Furthermore, these functions are related via:*

$$Q_\alpha^\pi(s, a) = r(s, a) + \gamma \frac{\beta^{-1}}{1 - \alpha} \log \mathbb{E}_{s' \sim p(s'|s,a)} \left[ e^{\beta(1-\alpha)V_\alpha^\pi(s')} \right] \tag{20}$$

$$V_\alpha^\pi(s) = \frac{\beta^{-1}}{1 - \alpha} \log \mathbb{E}_{a \sim \pi(a|s)} \left[ \left( \frac{e^{\beta Q_\alpha^\pi(s,a)} \pi_b(a|s)}{\pi(a|s)} \right)^{1-\alpha} \right] \tag{21}$$

The proof of this theorem is given in App. A.2.

Having established a theoretical basis for policy evaluation, we now turn to policy improvement. Our task is to generalise the notion of greedy action selection to the Rényi regularised setting. We wish to know, given our current policy's action-value function, how to define a new policy which has a greater action-value function. Equation (21) can be alternatively written as

$$V_\alpha^\pi(s) = V^*(s) - \beta^{-1} D_\alpha \left( \pi(a|s) || \pi^*(a|s) \right), \tag{22}$$

where

$$\pi^*(a|s) = \pi_b(a|s) \exp(\beta(Q_\alpha^\pi(s, a) - V^*(s))) \tag{23}$$

is the Boltzmann policy with respect to $Q_\alpha^\pi$, and

$$V^*(s) = \beta^{-1} \log \mathbb{E}_{a \sim \pi_b(a|s)} \left[ e^{\beta Q_\alpha^\pi(s,a)} \right] \tag{24}$$

is the appropriate log-normalisation factor. From this we can see that improving the value of a state is equivalent to reducing the $\alpha$-Rényi divergence between the policy at that state and the Boltzman policy. We capture this in the following theorem:

**Theorem 2** ($\alpha$-soft policy improvement). *Consider a finite MDP, i.e., $|\mathcal{S} \times \mathcal{A}| < \infty$. Then for any policy $\pi$, let $\pi^*$ be the corresponding Boltzmann policy, given by Eq. (23). If $\pi_{\text{new}}$ satisfies*

$$D_\alpha \left( \pi_{\text{new}}(a'|s') || \pi^*(a'|s') \right) \leq D_\alpha \left( \pi(a'|s') || \pi^*(a'|s') \right), \forall s' \in \mathcal{S}, \tag{25}$$

*then $Q_\alpha^{\pi_{\text{new}}} \geq Q_\alpha^\pi$. Moreover, for any state-action pair $(s, a)$ which has a non-zero probability of transitioning into a state $s'$ at which the inequality in Eq. (25) is strict, we have that $Q_\alpha^{\pi_{\text{new}}}(s, a) > Q_\alpha^\pi(s, a)$.*

The proof of this theorem is given in App. A.3.

Theorem 2 tells us how we can improve our policy, namely by decreasing the $\alpha$-Rényi divergence with the Boltzmann policy at every state. Our last result concerns the generalisation of the value iteration procedure to this new setting. This will allow us to formulate an off-policy algorithm analogous to Q-learning (Watkins & Dayan, 1992; Mnih et al., 2015). This is done by updating action-values $Q$ according to a policy that is Boltzmann with respect to the current $Q$ function. Equivalently, we set the next state-value function in Eq. (20) to be $V^*(s) = \beta^{-1} \mathbb{E}_{a \sim \pi_b(a|s)} [\exp(\beta Q(s, a)]$, as in Eq. (24). Doing so gives us an update purely in terms of action-value functions - the $\alpha$-*soft Bellman optimality operator*,

$$[\mathcal{B}_\alpha^* Q](s, a) = r(s, a) + \gamma \frac{\beta^{-1}}{1 - \alpha} \log \mathbb{E}_{s' \sim p(s'|s,a)} \left[ \mathbb{E}_{a' \sim \pi_b(a'|s')} \left[ e^{\beta Q(s',a')} \right]^{1-\alpha} \right] \tag{26}$$

**Theorem 3** ($\alpha$-soft value iteration). *Consider a finite MDP, i.e., $|\mathcal{S} \times \mathcal{A}| < \infty$. Then for any $\gamma \in (0, 1)$, the $\alpha$-soft Bellman optimality operator $\mathcal{B}_\alpha^*$ is a contraction mapping in the $\ell^\infty$ norm with contraction modulus $\gamma$. Accordingly, there exists a unique fixed point, which we call the $\alpha$-soft optimal action-value function, and denote by $Q_\alpha^*(s, a)$, to which any sequence of iterates converges in the $\ell^\infty$ norm. Furthermore, we have that:*

$$Q_\alpha^*(s, a) = \sup_\pi Q_\alpha^\pi(s, a) \tag{27}$$

The proof of this theorem is given in App. A.4.

Having now established key theoretical results for the Rényi regularised RL setting, we turn our attention to practical algorithms for the non-tabular case.

## 4 FROM THEORY TO ALGORITHMS

We devise two novel algorithms for the discrete action-space setting, $\alpha$-Soft Actor-Critic ($\alpha$-SAC) and $\alpha$-Soft Q-Learning ($\alpha$-SQL). $\alpha$-SAC and $\alpha$-SQL both make use of a collection of action-value functions $Q(s, a; \phi_k), k = 1, \ldots, K$, which take in states and output values for each action. We will take the minimum over these when computing action-values, to compensate for value over-optimism (Fujimoto et al., 2018). Additionally, $\alpha$-SAC makes use of a parametric policy network $\pi(a|s; \theta)$, which takes in states and outputs probabilities over actions. We will also use delayed action value-functions, $Q(s, a; \phi_k^-)$, whose parameters $\phi_k^-$ are synchronised with $\phi_k$ after a fixed number of update steps. Both $\alpha$-SAC and $\alpha$-SQL are off-policy, and make use of a finite capacity, first-in-last-out (FILO) memory replay buffer into which (state, action, reward, next state, done) transitions, $(s, a, r, s', d)$ are loaded and then resampled.

Full pseudocode for both algorithms is found in App. B.

### 4.1 VALUE LEARNING

To fit the value function parameters $\phi_k$ we obtain gradient from the mean-squared error:

$$\mathcal{L}(\phi_k) = \frac{1}{N} \sum_{i=1}^{N} (y_i - Q(s_i, a_i; \phi_k))^2 \tag{28}$$

where $y_i = y(r_i, s_i', d_i)$ are regression targets, and the sum is taken over a mini-batch of $N$ transitions sampled from the memory replay buffer, $\{(s_i, a_i, r_i, s_i', d_i)\}_{i=1}^{N}$. The regression targets are derived from either the $\alpha$-soft Bellman operator, Eq. (19), in the case of $\alpha$-SAC, or the $\alpha$-soft Bellman optimality operator, Eq. (26), in the case of $\alpha$-SQL. Note that both of these updates involve an expectation over transitions. We will therefore concentrate only on the case of deterministic environments, for which a single sample suffices for transition dynamics. The $\alpha$-SAC regression targets are given by:

$$y_i = r_i + \gamma \frac{\beta^{-1}}{1 - \alpha} \log \mathbb{E}_{a_i' \sim \pi(a_i'|s_i'; \theta)} \left[ \left( \frac{e^{\beta \min_k Q(s_i', a_i'; \phi_k^-)} \pi_b(a_i'|s_i')}{\pi(a_i'|s_i'; \theta)} \right)^{1-\alpha} \right], \tag{29}$$

where the expectation is computed by summing over the finite collection of actions. The minimum over action-value functions is applied to combat value overoptimism (Fujimoto et al., 2018). To find the $\alpha$-SQL regression targets, we note that, for a deterministic environment, the outer expectation over next states in the $\alpha$-soft Bellman optimality operator, Eq. (26), collapses to a single sample. The regression targets are therefore given by:

$$y_i = r_i + \gamma \beta^{-1} \log \mathbb{E}_{a_i' \sim \pi_b(a_i'|s_i')} \left[ e^{\beta \min_k Q(s_i', a_i'; \phi_k^-)} \right]. \tag{30}$$

Note that this is independent of $\alpha$. Thus, the parameter $\alpha$ only effects policy learning in $\alpha$-SQL.

### 4.2 POLICY LEARNING

Policy gradients for $\alpha$-SAC are obtained by performing ascent on the average value of states sampled from the memory replay buffer, as given by Eq. (21):

$$J(\theta) = \frac{1}{N} \sum_{i=1}^{N} \frac{\beta^{-1}}{1 - \alpha} \log \mathbb{E}_{a_i \sim \pi(a_i|s_i; \theta)} \left[ \left( \frac{e^{\beta \min_k Q(s_i, a_i; \phi_k)} \pi_b(a_i|s_i)}{\pi(a_i|s_i; \theta)} \right)^{1-\alpha} \right]. \tag{31}$$

Once again, the expectation is computed by summing over the available actions. By Theorem 2, we know that if the action-value function were correct, then increasing this objective at every state yields a strictly better policy. We settle instead for sampling states from the memory replay buffer and doing ascent at those states. In light of Eq. (22), we can alternatively interpret ascent on this objective as minimising the $\alpha$-Rényi divergence between the policy network and the Boltzmann policy given by the current action-value function at a sample of states.

For $\alpha$-SQL, we simply take our policy to be Boltzmann with respect to the current action-value function, *i.e.*,

$$\pi(a|s) = \frac{\pi_b(a|s) e^{\beta \min_k Q(s, a; \phi_k)}}{\sum_{\tilde{a}} \pi_b(\tilde{a}|s) e^{\beta \min_k Q(s, \tilde{a}; \phi_k)}} \tag{32}$$

### 4.3 AUTOMATIC REWARD SCALING ADJUSTMENT

Finally, we consider a mechanism which automatically adjusts the parameter $\beta$. We denote by $\bar{D}$ an average target divergence that we wish to maintain. For both $\alpha$-SAC and $\alpha$-SQL, we adjust the inverse temperature $\beta^{-1}$ by doing descent on the following loss:

$$\mathcal{L}(\beta^{-1}) = \beta^{-1}\left(\bar{D} - \frac{1}{N}\sum_{i=1}^{N} D(s_i)\right) \tag{33}$$

where $D(s_i)$ is the Rényi divergence at state $s_i$, which for $\alpha$-SAC is computed as

$$D(s) = \frac{1}{\alpha-1}\log \mathbb{E}_{a\sim\pi(a|s;\theta)}\left[\left(\frac{\pi(a|s;\theta)}{\pi_b(a|s)}\right)^{\alpha-1}\right]. \tag{34}$$

For $\alpha$-SQL, we have an analytic form of the $\alpha$-Rényi divergence, expressed in terms of the action-value function:

$$D(s) = \frac{1}{\alpha-1}\left(\log \mathbb{E}_{a\sim\pi_b(a|s)}\left[e^{\alpha\beta\min_k Q(s,a;\phi_k)}\right] - \alpha\log \mathbb{E}_{a\sim\pi_b(a|s)}\left[e^{\beta\min_k Q(s,a;\phi_k)}\right]\right). \tag{35}$$

This procedure mimics the automatic temperature adjustment mechanism used in SAC Haarnoja et al. (2019) and SAC-Discrete (Christodoulou, 2019).

## 5 RESULTS

We test $\alpha$-SAC and $\alpha$-SQL on four Atari environments in the Gymnasium package (Mnih et al., 2015; Brockman et al., 2016; Towers et al., 2024) - Qbert, Ms Pacman, Assault, and Space Invaders . For all of these we use the version 5 environment. As a baseline, we compare to our re-implementation of SAC-Discrete (Christodoulou, 2019). We examine the behaviour of $\alpha$-SAC and $\alpha$-SQL for two values of $\alpha$ below and above 1 (which corresponds to the KL-regularised case): $\alpha = 0.95$ and $\alpha = 1.05$, respectively. To allow a faithful comparison, the only hyperparameters we tune are $\alpha$ and the target divergence $\bar{D}$, leaving all other hyperparameters identical to those used by SAC-Discrete (Christodoulou, 2019). Note that the target divergence $\bar{D}$ is not tuned individually for each environment, but rather set to be a multiple of the $\log(|A|)$ where, $|A|$ is the number of actions in the environment. The network architecture, training hyperparameters, and additional pre-processing details can be found in App. C. For each environment we average results over 10 random seeds. We train for a total of $500,000$ environment steps. Every $4000$ environment steps, we evaluate the policy by averaging the empirical return over 5 episodes.

In Fig. 2 we compare the performance of our algorithms $\alpha$-SAC and $\alpha$-SQL to SAC-D across four environments. We see that our methods are able to learn in all four environments. The biggest different in performance is between the $\alpha$-SQL methods and the SAC methods. We see that $\alpha$-SQL consistently achieves higher returns than SAC in all four environments. We additionally note that $\alpha$-SQL has a lower compute cost compared to the SAC algorithms, since it uses only action-value networks, and not a policy network. Among the SAC methods, we see that $\alpha$-SAC is competitive with SAC-D in both Qbert and Space Invaders, and is able to outperform SAC-D for both Ms Pacman and Assault. Moreover, in those environments, we see that $\alpha = 1.05$ tends to outperform $\alpha = 0.95$. The value of $\alpha$ appears less importance for $\alpha$-SQL; this is likely because in $\alpha$-SQL, $\alpha$ affects only temperature adjustment, but not regression target formation.

## 6 DISCUSSION

Our work builds upon and extends important foundational work in RL which relates regularised RL to approximate probabilistic inference. Our main goal in this paper has been to theoretically illustrate the potential of this framework for developing novel RL algorithms. Indeed, unlike previous work, we take probabilistic inference as the starting point for new development, rather than reward regularisation. We hope that future work can extend this core idea by using other variational approximate inference methods, such as importance weighted variational inference (Burda et al., 2016) and f-divergence variational inference (Wan et al., 2020), in the RL setting.

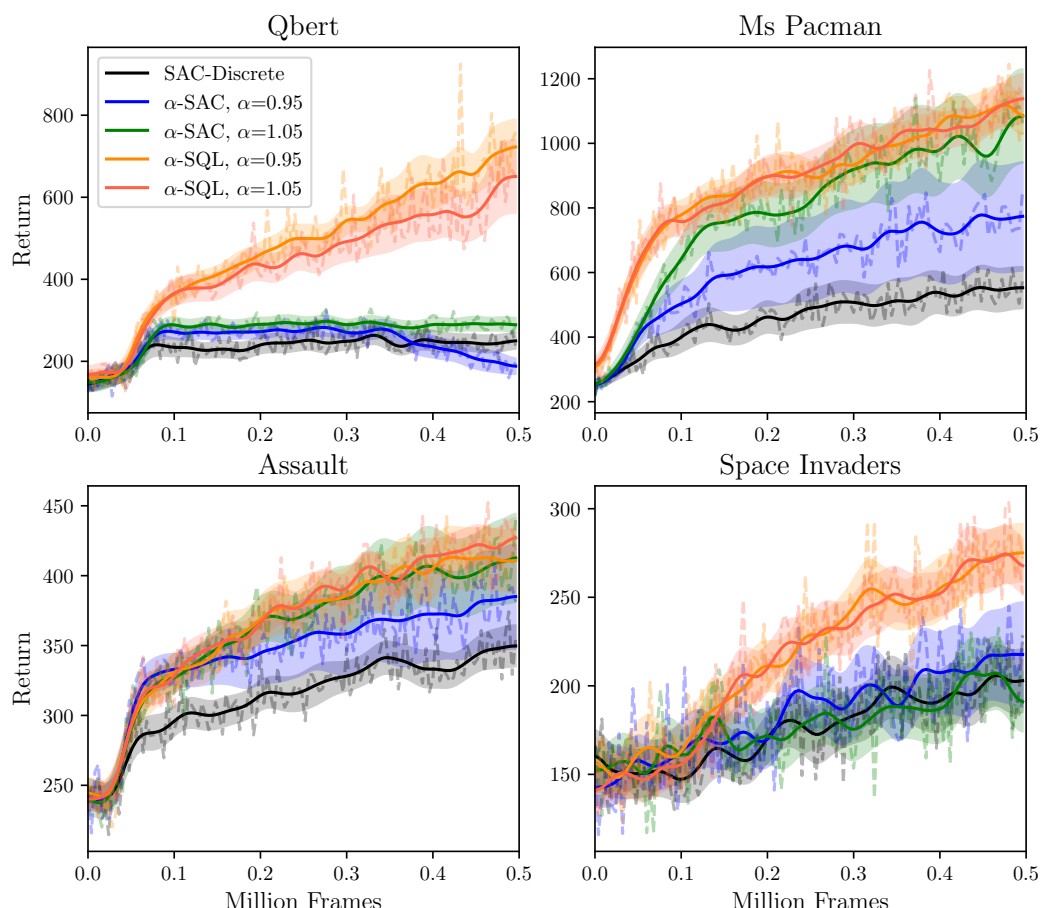

Figure 2: **The performance of $\alpha$-SAC and $\alpha$-SQL on four Atari environments**. In each panel, the dashed lines give the empirical returns averaged over 10 random seeds. The solid lines are give smoothed versions of the returns, obtained by Gaussian smoothing. The shaded area indicates the smoothed returns $\pm$ the smoothed standard error.

For our practical algorithms, we have considered only regularisation towards a (potentially unnormalised) uniform base policy. However, many applications of KL-regularisation use a non-uniform base policy, for example, the behavioural policy (or an approximation to it) in offline RL (Figueiredo Prudencio et al., 2024), or a pre-trained policy in Reinforcement Learning from Human Feedback applied to language models (Zheng et al., 2023). These situations provide additional possible uses for Rényi regularised RL. Note that, following the discussion in Sec. 2.3, varying $\alpha$ varies the extent to which we penalise our new policy for generating trajectories that have low (or zero) probability under the old policy. Thus, by varying $\alpha$ independently from the target divergence value, we can control not only the strength of regularisation but also its form, and in particular how severely it penalises trajectories which are out-of-distribution with respect to the base policy.

Unlike in the original formulation of $\alpha$-Rényi variational inference, we have only formulated the $\alpha$-SAC and $\alpha$-SQL algorithms for the fixed $\alpha$ values. We hope that future work may extend our formalism to include automatic adjustments of $\alpha$ according to some other criterion.

In this paper, we have only investigated Rényi regularised algorithms for the discrete action setting, rather than for continuous action spaces. In the continuous setting, the regression targets used for learning action-values (Eq. (29) and Eq. (30)) must make use of Monte-Carlo approximations for the expectations over actions. Initial experiments revealed that, although the algorithm was able to learn, the number of samples necessary for generating faithful approximations made these method prohibitively costly, and so we decided to focus on the discrete setting. It remains to be seen if other

variance reduction techniques could be used to mitigate this problem and make this method viable for the continuous action setting.

CONCLUSION

We have extended the "RL as probabilistic inference" framework by considering an alternative to the approximate inference objective which uses $\alpha$-Rényi variational inference. This lead us to formulate the $\alpha$-Rényi RL objective. This objective generates its own set of Bellman recursion relationships and backup operators, for which we provided theoretical results for both policy evaluation and improvement. We then leveraged these results in the formulation of two new algorithms, $\alpha$-SAC and $\alpha$-SQL. We gave concrete implementations of these methods in the case of discrete action-spaces, and demonstrated that they perform favourably against their KL-regularised counterpart, SAC-Discrete.

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

## A Proofs of theoretical results

### A.1 Bellman recursion relationship for the $\alpha$-soft state-value function

$$e^{\beta(1-\alpha)V_\alpha^\pi(s_0)} = \mathbb{E}\left[\prod_{t=0}^{T-1}\left(\frac{e^{\beta r(s_t,a_t)}\pi_b(a_t|s_t)}{\pi(a_t|s_t)}\right)^{(1-\alpha)}\Bigg| s_0\right]$$

$$= \mathbb{E}\left[\left(\frac{e^{\beta r(s_0,a_0)}\pi_b(a_0|s_0)}{\pi(a_0|s_0)}\right)^{1-\alpha}\prod_{t=1}^{T-1}\left(\frac{e^{\beta r(s_t,a_t)}\pi_b(a_t|s_t)}{\pi(a_t|s_t)}\right)^{(1-\alpha)}\Bigg| s_0\right]$$

$$= \mathbb{E}\left[\left(\frac{e^{\beta r(s_0,a_0)}\pi_b(a_0|s_0)}{\pi(a_0|s_0)}\right)^{1-\alpha}\mathbb{E}\left[\prod_{t=1}^{T-1}\left(\frac{e^{\beta r(s_t,a_t)}\pi_b(a_t|s_t)}{\pi(a_t|s_t)}\right)^{1-\alpha}\Bigg| s_1\right]\Bigg| s_0\right]$$

$$= \mathbb{E}\left[\left(\frac{e^{\beta r(s_0,a_0)}\pi_b(a_0|s_0))}{\pi(a_0|s_0)}\right)^{1-\alpha}e^{\beta(1-\alpha)V_\alpha^\pi(s_1)}\Bigg| s_0\right]$$

$$= \mathbb{E}\left[\left(\frac{e^{\beta r(s_0,a_0)}\pi_b(a_0|s_0)}{\pi(a_0|s_0)}\right)^{1-\alpha}\mathbb{E}\left[e^{\beta(1-\alpha)V_\alpha^\pi(s_1)}\Big| s_0,a_0\right]\Bigg| s_0\right]$$

### A.2 Proof of the $\alpha$-soft policy evaluation theorem

We will show that the $\alpha$-soft Bellman operator defined by Eq. (18) is a contraction mapping in the $\ell^\infty$ norm over action-value functions. We begin by defining the density $\hat{p}(s',a'|s,a)$ given by:

$$\hat{p}(s',a'|s,a) = \frac{1}{Z(s,a)}\pi(a'|s')p(s'|s,a)\left(\frac{\pi_b(a'|s')}{\pi(a'|s')}\right)^{1-\alpha} \tag{36}$$

where $Z(s,a) > 0$ is a normalisation constant. We will additionally reduce notational clutter by letting $\bar{Q} := \beta(1-\alpha)Q$. Then we can re-write the $\alpha$-soft Bellman operator acting on action-value functions as:

$$\left[\mathcal{B}_\alpha^\pi \bar{Q}\right](s,a) := \bar{r}(s,a) + \log Z(s,a) + \gamma\log\mathbb{E}_{a',s'\sim\hat{p}(s',a'|s,a)}\left[e^{\bar{Q}(s',a')}\right] \tag{37}$$

We will show that this is a contraction map by contradiction. Suppose not. Then for some choice of $Q(s,a)$ and $U(s,a)$, we can say that

$$\sup_{s,a\in\mathcal{S}\times\mathcal{A}}|[\mathcal{B}_\alpha^\pi Q](s,a) - [\mathcal{B}_\alpha^\pi U](s,a)| > \gamma\sup_{s',a'\in\mathcal{S}\times\mathcal{A}}|Q(s',a') - U(s',a')| \tag{38}$$

In particular, (w.l.o.g., by exchanging $Q$ and $U$), we can say that:

$$\left[\mathcal{B}_\alpha^\pi \bar{Q}\right](s,a) - \left[\mathcal{B}_\alpha^\pi \bar{U}\right](s,a) > \gamma\bar{Q}(s',a') - \gamma\bar{U}(s',a'), \; \forall(s',a')\in\mathcal{S}\times\mathcal{A} \tag{39}$$

We can now apply Eq. (37) to say that:

$$\gamma\log\mathbb{E}_{a',s'\sim\hat{p}(s',a'|s,a)}\left[e^{\bar{Q}(s',a')}\right] - \gamma\log\mathbb{E}_{a',s'\sim\hat{p}(s',a'|s,a)}\left[e^{\bar{U}(s',a')}\right] >$$
$$\gamma\bar{Q}(s',a') - \gamma\bar{U}(s',a'), \; \forall(s',a')\in\mathcal{S}\times\mathcal{A} \tag{40}$$

We now divide through by the discount factor $\gamma$, and apply $\exp(\bullet)$ to both sides. As this is strictly increasing, this implies that:

$$\mathbb{E}_{a',s'\sim\hat{p}(s',a'|s,a)}\left[e^{\bar{Q}(s',a')}\right]e^{\bar{U}(s',a')} >$$
$$\mathbb{E}_{a',s'\sim\hat{p}(s',a'|s,a)}\left[e^{\bar{U}(s',a')}\right]e^{\bar{Q}(s',a')}, \; \forall(s',a')\in\mathcal{S}\times\mathcal{A} \tag{41}$$

We now take expectations of both sides with respect to $\hat{p}(s',a'|s,a)$ to arrive at a contradiction.

The proof strategy for the case of state-value functions is very similar. We first define the modified density

$$\hat{p}(a|s) = \frac{1}{Z(s)} \left( \frac{e^{\beta r(s,a)} \pi_b(a|s)}{\pi(a|s)} \right)^{1-\alpha} \pi(a|s), \tag{42}$$

where $Z(s) > 0$ is a normalisation constant. Then we can re-express the $\alpha$-soft Bellman operator defined in Eq. (18) using $\hat{p}(a|s)$ as follows:

$$\left[ \mathcal{B}_\alpha^\pi \bar{V} \right](s) = \log Z(s) + \log \mathbb{E}_{a \sim \hat{p}(a|s)} \left[ \mathbb{E} \left[ e^{\bar{V}(s')} | s, a \right]^\gamma \right] \tag{43}$$

We will once again argue by contradiction. Then for some choice of $\bar{V}$ and $\bar{U}$, and $s \in \mathcal{S}$, we have

$$\left[ \mathcal{B}_\alpha^\pi \bar{V} \right](s) - \left[ \mathcal{B}_\alpha^\pi \bar{U} \right](s) > \gamma \bar{V}(s') - \gamma \bar{U}(s'), \ \forall s' \in \mathcal{S} \tag{44}$$

We divide both sides by $\gamma$ and use Eq. (43) to obtain:

$$\log \mathbb{E}_{a \sim \hat{p}(a|s)} \left[ \mathbb{E} \left[ e^{\bar{V}(s')} | s, a \right]^\gamma \right]^{1/\gamma} - \log \mathbb{E}_{a \sim \hat{p}(a|s)} \left[ \mathbb{E} \left[ e^{\bar{U}(s')} | s, a \right]^\gamma \right]^{1/\gamma} >$$
$$\bar{V}(s') - \bar{U}(s'), \ \forall s' \in \mathcal{S} \tag{45}$$

We now apply $\exp(\bullet)$ to both sides, and then take expectations with respect to $p(s'|s, a)$ to obtain that:

$$\mathbb{E}_{a \sim \hat{p}(a|s)} \left[ \mathbb{E} \left[ e^{\bar{V}(s')} | s, a \right]^\gamma \right]^{1/\gamma} \mathbb{E} \left[ e^{\bar{U}(s')} | s, a \right] >$$
$$\mathbb{E}_{a \sim \hat{p}(a|s)} \left[ \mathbb{E} \left[ e^{\bar{U}(s')} | s, a \right]^\gamma \right]^{1/\gamma} \mathbb{E} \left[ e^{\bar{V}(s')} | s, a \right], \forall a \in \mathcal{A} \tag{46}$$

To complete the proof, we exploit strict monotonicity to raise both sides to the power of $\gamma$, and then take expectations over $\hat{p}(a|s)$ to arrive at a contradiction. This completes the first half of the proof, and shows that the $\alpha$-soft state- and action- value functions are indeed well-defined as unique fixed points of the corresponding Bellman operators. Furthermore, we have that any sequence of iterates converges in the $\ell^\infty$-norm to these fixed points.

For the second half of the proof, we establish recursion relationships that holds between the $\alpha$-soft state- and action-value functions, $V_\alpha^\pi(s)$ and $Q_\alpha^\pi$. We will start by showing Eq. (20). Let us define:

$$Q(s,a) = r(s,a) + \gamma \frac{\beta^{-1}}{1-\alpha} \log \mathbb{E}_{s' \sim p(s'|s,a)} \left[ e^{\beta(1-\alpha)V_\theta^\alpha(s')} \right]. \tag{47}$$

We will show that this is a fixed point of the $\alpha$-soft Bellman operator over action-value functions, and thus is equal to $Q_\alpha^\pi(s,a)$. We proceed as follows:

$$[\mathcal{B}_\alpha^\pi Q](s,a) = r(s,a) + \gamma \frac{\beta^{-1}}{1-\alpha} \log \mathbb{E} \left[ \left( \frac{e^{\beta Q(s',a')} \pi_b(a'|s')}{\pi(a'|s')} \right)^{1-\alpha} \Bigg| s, a \right]$$

$$= r(s,a) + \gamma \frac{\beta^{-1}}{1-\alpha} \log \mathbb{E} \left[ \left( \frac{\pi_b(a'|s')}{\pi(a'|s')} \right)^{1-\alpha} e^{\beta(1-\alpha)Q(s',a')} \Bigg| s, a \right]$$

$$= r(s,a) + \gamma \frac{\beta^{-1}}{1-\alpha} \log \mathbb{E} \left[ \left( \frac{\pi_b(a'|s')}{\pi(a'|s')} \right)^{1-\alpha} e^{\beta(1-\alpha)r(s',a')} \mathbb{E} \left[ e^{\beta(1-\alpha)V_\alpha^\pi(s'')} \Big| s', a' \right]^\gamma \Bigg| s, a \right]$$

$$= r(s,a) + \gamma \frac{\beta^{-1}}{1-\alpha} \log \mathbb{E} \left[ \mathbb{E} \left[ \left( \frac{e^{\beta r(s',a')} \pi_b(a'|s')}{\pi(a'|s')} \right)^{1-\alpha} \mathbb{E} \left[ e^{\beta(1-\alpha)V_\alpha^\pi(s'')} \Big| s', a' \right]^\gamma \Bigg| s' \right] \Bigg| s, a \right]$$

$$= r(s,a) + \gamma \frac{\beta^{-1}}{1-\alpha} \log \mathbb{E} \left[ e^{\beta(1-\alpha)V_\alpha^\pi(s')} \Big| s, a \right]$$

$$= Q(s,a)$$

as required. We will now establish the converse relationship, given by Eq. (21). To do this, we will define:

$$V(s) = \frac{\beta^{-1}}{1-\alpha} \log \mathbb{E}_{a \sim \pi(a|s)} \left[ \left( \frac{e^{\beta Q_\alpha^\pi(s,a)} \pi_b(a|s)}{\pi(a|s)} \right)^{1-\alpha} \right]. \tag{48}$$

As before, we show that $V$ is a fixed point of the $\alpha$-soft Bellman operator over state-value functions, and is thus equal to $V_\alpha^\pi(s)$. First, note that:

$$
e^{\beta(1-\alpha)r(s,a)}\mathbb{E}\left[e^{\beta(1-\alpha)V(s')}\Big|s,a\right]^\gamma = e^{\beta(1-\alpha)r(s,a)}\mathbb{E}\left[\mathbb{E}\left[\left(\frac{e^{\beta Q_\alpha^\pi(s',a')}\pi_b(a'|s')}{\pi(a'|s')}\right)^{1-\alpha}\Bigg|s'\right]\Bigg|s,a\right]^\gamma
$$

$$
= e^{\beta(1-\alpha)r(s,a)}\mathbb{E}\left[\left(\frac{e^{\beta Q_\alpha^\pi(s',a')}\pi_b(a'|s')}{\pi(a'|s')}\right)^{1-\alpha}\Bigg|s,a\right]^\gamma
$$

$$
= e^{\beta(1-\alpha)Q_\alpha^\pi(s,a)}.
$$

We now apply this result to simplify the $\alpha$-soft Bellman operator applied to $V$:

$$
\left[\mathcal{B}_\alpha^\pi V\right](s) = \frac{\beta^{-1}}{1-\alpha}\log\mathbb{E}\left[\left(\frac{e^{\beta r(s,a)}\pi_b(a|s)}{\pi(a|s)}\right)^{1-\alpha}\mathbb{E}\left[e^{\beta(1-\alpha)V(s')}\Big|s,a\right]^\gamma\Bigg|s\right]
$$

$$
= \frac{\beta^{-1}}{1-\alpha}\log\mathbb{E}\left[\left(\frac{\pi_b(a|s)}{\pi(a|s)}\right)^{1-\alpha}e^{\beta(1-\alpha)Q_\alpha^\pi(s,a)}\Bigg|s\right]
$$

$$
= \frac{\beta^{-1}}{1-\alpha}\log\mathbb{E}\left[\left(\frac{e^{\beta Q_\alpha^\pi(s,a)}\pi_b(a|s)}{\pi(a|s)}\right)^{1-\alpha}\Bigg|s\right]
$$

$$
= V(s).
$$

Therefore $V(s) = V_\alpha^\pi(s)$, as claimed.

### A.3 POLICY IMPROVEMENT

Here we prove the $\alpha$-soft policy improvement theorem. We will first prove a more general lemma, before turning to the main result.

**Lemma 1.** *Let $Q(s,a)$ be an action-value function with corresponding Boltzmann policy $\pi^*(a'|s') = \pi_b(a'|s')\exp(\beta(Q(s',a') - V^*(s')))$, where $V^*$ serves to normalise the density. Then consider any two policies $\pi_1$ and $\pi_2$, which satisfy:*

$$
D_\alpha(\pi_1(a'|s')||\pi^*(a'|s')) \leq D_\alpha(\pi_2(a'|s')||\pi^*(a'|s')) \ \forall s' \in \mathcal{S} \tag{49}
$$

*then $\left[\mathcal{B}_\alpha^{\pi_1}Q\right](s,a) \geq \left[\mathcal{B}_\alpha^{\pi_2}Q\right](s,a)$, with strict inequality at any $(s,a)$ which has non-zero probability of transitioning into $s'$ at which the inequality is strict in Eq. (49).*

*Proof.* First, note that the $\alpha$-soft Bellman operator acting on action-value functions can be re-expressed as:

$$
\left[\mathcal{B}_\alpha^\pi Q\right](s,a) = r(s,a) + \gamma\frac{\beta^{-1}}{1-\alpha}\log\mathbb{E}_{s'\sim p(s'|s,a)}\left[e^{(1-\alpha)[\beta V^*(s')-D_\alpha(\pi(a'|s')||\pi^*(a'|s'))]}\right] \tag{50}
$$

From here, the proof is relatively straightforward. In the case $1 - \alpha > 0$, we note that it suffices to show that

$$
e^{\gamma^{-1}\beta(1-\alpha)[\mathcal{B}_\alpha^{\pi_1}Q](s,a)} - e^{\gamma^{-1}\beta(1-\alpha)[\mathcal{B}_\alpha^{\pi_2}Q](s,a)} \geq 0 \tag{51}
$$

But, using Eq. (50), this is equivalent to

$$
\mathbb{E}_{s'\sim p(s'|s,a)}\left[e^{(1-\alpha)[\beta V^*(s')-D_\alpha(\pi_1(a'|s')||\pi^*(a'|s'))]}\right] \geq
$$

$$
\mathbb{E}_{s'\sim p(s'|s,a)}\left[e^{(1-\alpha)[\beta V^*(s')-D_\alpha(\pi_2(a'|s')||\pi^*(a'|s'))]}\right] \tag{52}
$$

which holds because of Eq. (49), with strict inequality if there is a non-zero probability of $s, a$ transitioning to $s'$ where Eq. (49) holds strictly. The argument for $1 - \alpha < 0$ is almost identical, after exchanging $\pi_1$ and $\pi_2$ in both Eq. (51) and Eq. (52). $\square$

The stated result in Theorem 2 now follows almost immediately from Lemma 1. Take $Q = Q_\alpha^\pi$, $\pi_1 = \pi$, and $\pi_2 = \pi_{\text{new}}$ satisfying Eq. (25). Then we have that

$$\left[\mathcal{B}_\alpha^{\pi_{\text{new}}} Q_\alpha^\pi\right](s,a) \geq \left[\mathcal{B}_\alpha^\pi Q_\alpha^\pi\right](s,a) = Q_\alpha^\pi(s,a) \tag{53}$$

with strict inequality whenever $s, a$ has a non-zero probability of transitioning into $s'$ where the inequality in Eq. (25) is strict. Note that the $\alpha$-soft Bellman operators are increasing, in the sense that if $Q \geq U$, then $\mathcal{B}_\alpha^{\pi_{\text{new}}} Q \geq \mathcal{B}_\alpha^{\pi_{\text{new}}} U$. We can thus argue that:

$$\left[\mathcal{B}_\alpha^{\pi_{\text{new}}}\right]^2 Q_\alpha^\pi \geq \mathcal{B}_\alpha^{\pi_{\text{new}}} Q_\alpha^\pi \geq Q_\alpha^\pi. \tag{54}$$

Inductively, we see that for any $n \geq 0$, we must have that

$$\left[\left[\mathcal{B}_\alpha^{\pi_{\text{new}}}\right]^n Q_\alpha^\pi\right](s,a) \geq \left[\mathcal{B}_\alpha^{\pi_{\text{new}}} Q_\alpha^\pi\right](s,a) \geq Q_\alpha^\pi(s,a), \tag{55}$$

where the second inequality is strict inequality wherever $s, a$ has non-zero probability of transitioning into $s'$ at which the inequality in Eq. (25) is strict. We can now take the limit as $n \to \infty$ to achieve the desired result.

## A.4    PROOF OF THE $\alpha$-SOFT VALUE ITERATION THEOREM

Again, this theorem has two main parts - showing that $\mathcal{B}_\alpha^*$ is a contraction mapping (in the $\ell^\infty$ norm, with modulus $\gamma$), and then showing that the corresponding fixed point is optimal, in the sense of dominating all other action-value functions and being attained for some policy.

We begin with the contraction mapping proof, which is very similar in structure to the proofs in App. A.2. In particular, we assume a contradiction for some functions $Q$, $U$ at $(s,a)$. Then we can say that:

$$\gamma \frac{\beta^{-1}}{1-\alpha} \log \mathbb{E}\left[\mathbb{E}_{\pi_b}\left[e^{\beta Q(s',a')}\Big|s'\right]^{1-\alpha}\Big|s,a\right] - \gamma \frac{\beta^{-1}}{1-\alpha} \log \mathbb{E}\left[\mathbb{E}_{\pi_b}\left[e^{\beta Q(s',a')}\Big|s'\right]^{1-\alpha}\Big|s,a\right]$$
$$> \gamma Q(s',a') - \gamma U(s',a'), \ \forall (s',a') \in \mathcal{S} \times \mathcal{A} \tag{56}$$

We now divide through by $\gamma$ and multiply through by $\beta$. We apply $\exp(\bullet)$ to both sides of the equation, and take the expectation over $a' \sim \pi_b(a'|s')$ to obtain the following inequality:

$$\mathbb{E}\left[\mathbb{E}_{\pi_b}\left[e^{\beta Q(s',a')}\Big|s'\right]^{1-\alpha}\Big|s,a\right]^{\frac{1}{1-\alpha}} \mathbb{E}_{\pi_b}\left[e^{\beta U(s',a')}\Big|s'\right] >$$
$$\mathbb{E}\left[\mathbb{E}_{\pi_b}\left[e^{\beta U(s',a')}\Big|s'\right]^{1-\alpha}\Big|s,a\right]^{\frac{1}{1-\alpha}} \mathbb{E}_{\pi_b}\left[e^{\beta Q(s',a')}\Big|s'\right] \tag{57}$$

We now apply the mapping $x \mapsto x^{1-\alpha}$ to both sides. Note that, depending on whether $1 - \alpha > 0$ or $1 - \alpha < 0$, the mapping $x \mapsto x^{1-\alpha}$ will be either strictly increasing or decreasing respectively. In either case, we retain a strict inequality. We can now take expectations over $s' \sim p(s'|s,a)$ to arrive at a contradiction. This completes the contraction mapping portion of the proof, and establishes the existence of $Q_\alpha^*$.

We now turn to showing optimality. Let $\pi_{\text{opt}}$ be the policy which is everywhere Boltzmann with respect to $Q_\alpha^*$. Then by Eq. (22), we can see that $V_{\pi_{\text{opt}}}^\alpha = V^*$ everywhere. Then by Eq. (20) and Eq. (24), we have that $\mathcal{B}_{\pi_{\text{opt}}}^\alpha Q_{\pi_{\text{opt}}}^\alpha = \mathcal{B}_\alpha^* Q_{\pi_{\text{opt}}}^\alpha$. From this we can conclude that $Q_{\pi_{\text{opt}}}^\alpha$ is a fixed point of the $\alpha$-soft Bellman optimality operator, and therefore that $Q_\alpha^* = Q_{\pi_{\text{opt}}}^\alpha$. This shows that the supremum is attained somewhere. So it suffices only to show dominance.

To show the dominance relationship, we consider an arbitrary other policy $\tilde{\pi}$. We start by showing that $\mathcal{B}_\alpha^* Q \geq \mathcal{B}_\alpha^{\tilde{\pi}} Q$ for any action-value function $Q$. We let $\pi^*$ be Boltzmann with respect to $Q$. Then we apply Lemma 1 with $\pi_1 = \pi^*$ and $\pi_2 = \tilde{\pi}$, using the fact that the left-hand side of Eq. (49) is always zero and therefore the required inequality holds everywhere. This tells us that

$$\mathcal{B}_\alpha^* Q = \mathcal{B}_\alpha^{\pi^*} Q \geq \mathcal{B}_\alpha^{\tilde{\pi}} Q. \tag{58}$$

From this we can conclude that

$$Q_\alpha^* = \mathcal{B}_\alpha^* Q_\alpha^* \geq \mathcal{B}_\alpha^{\tilde{\pi}} Q_\alpha^* \tag{59}$$

Since $\mathcal{B}_\alpha^*$ is increasing as an operator (as can be seen by inspection of the definition), we can apply the result in Eq. (58) with $Q = \mathcal{B}_\alpha^{\tilde{\pi}} Q_\alpha^*$ to Eq. (59) to obtain

$$Q_\alpha^* = \mathcal{B}_\alpha^* Q_\alpha^* \geq \mathcal{B}_\alpha^* \mathcal{B}_\alpha^{\tilde{\pi}} Q_\alpha^* \geq \left[\mathcal{B}_\alpha^{\tilde{\pi}}\right]^2 Q_\alpha^* \tag{60}$$

Proceeding inductively, we have that $Q_\alpha^* \geq \left[\mathcal{B}_\alpha^{\tilde{\pi}}\right]^n Q_\alpha^*$ for $n \geq 0$. We can then take the limit as $n \to \infty$ and apply Theorem 1 to say that

$$Q_\alpha^* \geq Q_\alpha^{\tilde{\pi}}, \tag{61}$$

as required.

## B  PSEUDOCODE FOR THE $\alpha$-SAC AND $\alpha$-SQL ALGORITHMS

---

**Algorithm 1** The $\alpha$-SAC and $\alpha$-SQL algorithms

---

Initialise value networks, $Q(s, a; \phi_k), k = 1, \ldots, K$
Initialise parameters of target network, $\phi_k^- \leftarrow \phi_k$
For $\alpha$-SAC, Initialise policy network, $\pi(a|s; \theta)$
Initialise $\mathcal{D}$ as an empty FILO memory replay buffer with finite capacity
Load *learning starts* transitions sampled according to the uniform policy into $\mathcal{D}$
**for** each training step **do**
 ***Interact with the environment***
 **for** environment steps per update **do**
  Sample action $a \sim \pi(a|s; \theta)$ for $\alpha$-SAC and from $\pi(a|s)$ given by Eq. (32) for $\alpha$-SQL
  Get next state and reward, $s'(s, a), r(s, a)$
  $d \leftarrow 1$ if $s'$ is terminal, otherwise $d \leftarrow 0$
  Load the transition $(s, a, r, d, s')$ into memory replay buffer $\mathcal{D}$
  **if** $d = 1$ **then**
   Sample initial state $s \sim p_0(s)$
  **else**
   Set $s \leftarrow s'$
  **end if**
 **end for**
 ***Update parameters of the value networks***, $\phi_k$
 Sample $\{(s_i, a_i, d_i, r_i, s_i')\}_{i=1}^N$ from $\mathcal{D}$
 **for** $i = 1, \ldots, N$ **do**
  Form regression targets $y_i$ using Eq. (29) for $\alpha$-SAC and Eq. (30) for $\alpha$-SQL.
 **end for**
 Take gradients of losses $\mathcal{L}(\phi_k)$, Eq. (28)
 Update $\phi_k$ according to the Adam optimiser (or another optimiser).
 ***For $\alpha$-SAC, update parameters of the policy network***, $\theta$
 Take gradients on the policy objective $J(\theta)$, Eq. (31).
 Update $\theta$ according to the Adam optimiser (or another optimiser) in the ascent direction.
 ***Update the reward scaling parameter*** $\beta$
 Take gradient of the reward scaling loss, $\mathcal{L}(\beta^{-1})$, Eq. (33)
 Update $\beta$ according to the Adam optimiser (or another optimiser).
 ***Update target network parameters***
 **if** it's time to update the target networks **then**
  $\phi_k^- \leftarrow \phi_k$
 **end if**
**end for**

---

## C  HYPERPARAMETERS

### C.1  ENVIRONMENT PRE-PROCESSING

We perform standard (Mnih et al., 2015) pre-processing on the Atari environments as follows:

1. The Atari frames are converted to grayscale.

2. Grayscale inputs are scaled linearly from $[0, 255] \mapsto [0, 1]$

3. The frames are down-sampled to $84 \times 84$.

4. The observations sent to the agent are stacks of four consecutive frames.

5. During training, the rewards are binned based on their sign to $\{-1, 0, +1\}$. For evaluation rollouts, the reward clipping is removed.

## C.2 NETWORK ARCHITECTURE

For both the action-value and policy networks we use a convolutional neural network feature extractor, followed by an MLP which maps to a number of outputs each to the number of actions available for each state. ReLU non-linearities are applied between each linear layer. For the policy network, a final soft-max non-linearity is applied over the outputs, such that the output from the network is a categorical distribution over actions. The details of the architecture are shown in Table 1.

Table 1: Network architecture used for both the action-value and policy networks.

| Network hyperparameter | Value |
|---|---|
| Convolutional channels per layer | [32, 64, 64] |
| Convolutional kernel sizes per layer | [8, 4, 3] |
| Convolutional strides per layer | [4, 2, 1] |
| Convolutional padding per layer | [0, 0, 0] |
| MLP hidden layer units | [512, number of actions] |
| Non-linearity | ReLU |

## C.3 TRAINING HYPERPARAMETERS

Table 2: Hyperparameters used for the $\alpha$-SAC, $\alpha$-SQL, and SAC-Discrete algorithms

| Hyperparameter | Value |
|---|---|
| Batch size | 64 |
| Replay buffer capacity | 250000 |
| Discount factor $\gamma$ | 0.99 |
| Environment steps per network update | 4 |
| Learning rate | 0.0003 |
| Optimiser | Adam |
| Environment steps per target network update | 8000 |
| Learning starts | 20000 |
| Number of networks $K$ | 2 |
| **SAC-Discrete specific hyperparameters** | |
| Target policy entropy | $0.98 \times \log(|A|)$ |
| **$\alpha$-SAC specific hyperparameters** | |
| Regularisation policy $\pi_b$ | Unnormalised uniform, $\pi_b(a|s) = 1$ |
| Target $\alpha$-Divergence $D$ | $-0.9 \times \log(|A|)$ |
| **$\alpha$-SQL specific hyperparameters** | |
| Regularisation policy $\pi_b$ | Normalised uniform, $\pi_b(a|s) = 1/|A|$ |
| Target $\alpha$-Divergence $D$ | $0.1 \times \log(|A|)$ |

