# OpenReview forum: "Rényi Regularised Reinforcement Learning"
_ICLR.cc/2025/Conference — ICLR 2025 Conference Withdrawn Submission_

### Official Review · Reviewer_ztWE · 2024-10-18

**Soundness:** 1
**Presentation:** 2
**Contribution:** 2
**Rating:** 3
**Confidence:** 5

**Summary:**

This paper introduces a way to perform Rényi regularization of the behavior in reinforcement learning. Interestingly, it deviates from the common approach consisting in regularizing the reward/one-step policy, but rather regularize at the trajectory level, through the "control as inference" viewpoint, and then goes back to Bellman-like equations, on the transition level.

The authors present some relevant background, then the new Rényi regularized RL objective, introduce related concepts of value and Q-function, and corresponding Bellman operator. They state results about these operators, like their contractive aspects, a form of policy improvement result, and the facts that the fixed point of the introduced optimality Bellman operator dominates the Q-function of all other policies. They then propose a short empirical study on a few atari games with SAC-discrete as a a baseline, showing that the proposed $\alpha$-SQL consistently performs better than alternatives.

**Strengths:**

The overall proposed approach is really interesting and intriguing. The idea to start from the control as inference viewpoint to introduce regularization and go back to the MDP viewpoint is also very interesting, and makes sense. There is an effort to provide some theoretical evidences for the proposed approach. The paper is overall clearly written.

**Weaknesses:**

If the proposed idea is really interesting, the paper shows weaknesses both on the theoretical side and the empirical side, which prevents from supporting acceptance.

As a summary (detailed comment afterwards):
* on the theoretical side, a number of aspects could be clarified or better justified, and part of the proofs are wrong or at least incomplete (all those related to contraction). Even more importantly, in the discrete action and deterministic dynamics considered in practice (the only case for which a practical algorithm is presented), the proposed $\alpha$-SQL algorithm is nothing else that the well known soft-DQN, which is not acknowledged by the authors.
* the empirical study is far from being convincing. Considering discrete SAC as the sole baseline is really surprising (why not at least vanilla DQN?), the best results are obtained by $\alpha$-SQL which is nothing else than soft-DQN (not new), given the chosen benchmarks the number of training steps is far too low, there is no experiment showcasing advantages or properties of the proposed regularization (eg, exploration is used as an argument in the introduction, there could be other aspects).

# Theoretical aspect
## Clarity
* Eq (14), which is somehow the backbone of the paper, is justifed by replacing the ELBo in Eq (10) with the variational lower bound. This does not seem correct, how to show equivalence between (14) and something like $E_{\tau\sim q_\pi}[\beta G(\tau)] - D_\alpha(q_\pi||p)$? The issue being that $\ln$ and expectation are not commutative. Eq. (14) makes sense overall, but it could benefit from more justification/explanation.
* It would be interesting also to discuss what is being optimized (with either (14) or (15)), is it a different way of computing the same optimal policy as the classic control-as-inference entropy-based approach (as is the case for variational inference), or is it computing a different policy, and if so on what do they differ ?
* It would be helpful to bridge the definition of the value function, Eq (15), to the initial objective $L_\alpha(q_\pi)$, Eq (16). This objective is not the expected value (still log and expectation not commuting). Notably, if we compute the optimal policy through the value function as described later in the paper, does this policy optimize the objective of Eq (16)? It may be true, but it is not totally obvious.
* Given the definition of the value function in Eq (16), the Bellman recursion of Eq (17) is wrong, and the proof in Appx A.1 is wrong. It is a small typo, the lhs of the first equation in the proof should be $e^{(1-\alpha) V_\alpha^\pi(s)} = \dots$ and not $e^{\beta(1-\alpha) V_\alpha^\pi(s)} = \dots$, but this should be corrected (either by correcting Eq (17), which impacts directly all subsequent definitions and proofs, or by modifying the definition in Eq (15))

## Issues with proofs
There is a possible issue with the contraction proofs. In all these proofs, the authors use a contradiction argument, starting from
$$ \max_{z} |\[BQ\](z) - \[BU\](z) |> \gamma \max_z  | Q(z) - U(z) |$$
and then reaching a contradiction. To do so, they state that this equation implies, for some $z$ (maximizing the lhs), wlog by exchanging $Q$ and $U$, that
$$ \[BQ\](z) - \[BU\](z) > \gamma Q(z') - U(z'), \quad \forall z' $$
It is right that we do not need to consider the case where Q and U are swept, but what is missing is the following case:
$$ \[BQ\](z) - \[BU\](z) > \gamma U(z') - Q(z'), \quad \forall z' $$
The reason is that if the max on the lhs is reached, says for Q and z, that is $\max_{z''} | \[BQ\](z'') - \[BU\](z'') | = \[BQ\](z) - \[BU\](z)$, we may have that the max on the rhs is reached, not only for a different $z'$, but also for an asymmetric role of $Q$ and $U$ on that $z'$, that is $\max_{z''} \gamma | Q(z) - U(z) | = \gamma U(z') - Q(z')$.

Unfortunately, for this missing case, the same steps do not lead to a contradiction. The proof should be corrected or clarified.

## $\alpha$-SQL is soft-DQN

The only difference between the $\alpha$-soft Bellman optimality operator and the classic soft Bellman operator based on KL divergence is the place of the term $E_{s'\sim p(\cdot|s)}$, inside or outside the $\ln$ term. There are not equivalent, but this connexion could be discussed (notably what this implies). However, whenever the dynamics is deterministic, both become equivalent, which is not acknowledged by the authors. Given that the only practical case considered is when the dynamics is deterministic and the actions are discrete (due to the bias issue when using a sampled expectation within the $\ln$, only briefly discussed in the conclusion), this algorithm is not new.

# Empirical aspects

* the setting is not clear enough (eg, does the episode end when a life is lost or all of them, sticky actions or not, etc., all these aspects have a huge impact on the results)
* given the considered benchmarks (a few atari games), the number of training steps seems to be way too low (0.5M, while the standard is more like 50M), so it is hard to reach conclusions
* the achieved scores seem to be pretty low (eg compared to the baselines provided in the dopamine library), could be related to the too low number of training steps.
* the choice of the sole baseline considered is quite puzzling. Why discrete SAC? Why not considering a standard baseline such as DQN? Why not the relevant soft-DQN (it's there, as $\alpha$-SQL is nothing else than soft-DQN, but it is not stated in the paper)?
* the empirical study would benefit from ablations, for example on the effect of the automatic temperature scheduling, or for more various values of $\alpha$
* the study would also benefit from more controlled experiment to showcase the benefits of Rényi regularization, beyond the sole performance (eg, exploration was mentioned as the single motivation for doing regularization in the introduction, while there are other good reasons to do so, and no game is really hard exploration)
* why these games specifically?
* how where the target policy entropy chosen (in Appx C.3)?

Overall, the empirical study does not provide clear evidence of the interest of the proposed approach. This is even more true that what works best is in the end soft-DQN, which is not the more efficient approach, even among those exploiting regularization (eg, see the empirical study in "Munchausen reinforcement learning").

## Other comments
Here is a list of less important points, but that could help also improving the paper:
* the considered setting is finite horizon, which suggests looking for non-stationary optimal policies, this could be at least briefly discussed
* Eq (2), sounds strange to write an integral while discrete actions are considered in the whole paper
* l.142, $q(z) = p(x,z)$ should be $q(z) = p(z|x)$
* the Rényi divergence gives the KL one when $\alpha \rightarrow 1$, what would happen in you proposed loss/value functions?
* Beyond the concern of the value function defined Eq (17) already raised ($\beta$ factor), this Bellman recursion is not correct, because it ignores the finite aspect setting of the problem (the proof in Appx A.1 ignores the fact that the products involve different number of timesteps, and are thus time-dependent)
* Eq (22) and around, the notation $\pi^*$ is really misleading, as it suggests the optimal policy, while it is a soft-greedy policy, not necessarily optimal. Maybe that a notation like $\mathcal{Q^\pi_\alpha}$ or something similar would be clearer.
* l. 335, teh term $d$ appearing in the transition is not defined, what is it?
* l. 680, $\bar{r}$ is not defined (equal to $(1-\alpha)\beta r$, but could be explicitly stated)
* Eq (40), quite misleading to use the same notation within the expectation and outside (for the $(s',a')$ couples)
* Eq (47), $V_\theta^\alpha$ should be $V_\alpha^\pi$
* Eq. (50), would clarify to specify that it's from Eqs (20) and (22)

**Questions:**

Please address the questions and comments raised in the above detailed comments, notably:
* questions on clarity aspects
* please provide a complete proof for the contraction of operators (or explain why the current proof is ok, if it is the case)
* please comment on the fact that in the deterministic dynamics setting considered, the algorithm may not be new, as it seems to be soft-DQN in the end
* please provides answers to the empirical aspects question

---

### Official Review · Reviewer_LoNK · 2024-10-26

**Soundness:** 4
**Presentation:** 4
**Contribution:** 3
**Rating:** 6
**Confidence:** 4

**Summary:**

This work extends the control-as-inference framework from the case of MaxEnt/KL-regularized to Renyi-regularized RL. The authors provide theoretical results on the corresponding Bellman equations, contraction principle, policy evaluation & policy improvement. Additionally, the authors implement a discrete action version of their algorithm and verify its efficacy in several Atari environments.

**Strengths:**

1. From a mathematical standpoint, this work opens a generalization of the theory (across a new dimension, alpha) opening a bridge to other potential theoretical developments
2. The introductory discussion itself on control-as-inference and VI is useful addition to the literature.
3. The new theory yields non-trivial results, is well-founded and agrees with known results (alpha=1)
4. The experiments provided show potential for utility and adoption by the RL community.

**Weaknesses:**

1. Somewhat unclear theoretical advances:

    a. Are there new mathematical techniques being employed; or is it a reuse of old tricks. (OK if so, but worth stating to better understand the theory contribution... does the community gain new theoretical tools?). Can you highlight any new tools or show if the old tools were used in a novel way? Again, if it is all standard techniques (as it appears at first glance), that's fine but maybe can be stated upfront to understand the contribution better.

    b. Potential similarity to prior work on f divergences. (It is a bit unclear how this work relates / fits in the purview of the wider f-divergence framework, e.g. [1]). Can you please comment on this?


2. Experiments:

    a. (This is not a major concern, as I mostly view the work as theoretical) The use of Atari environments is nice, and demonstrates well the efficacy of the algorithms. However, I think there can be more work here. For example, can the authors give an ablation study with respect to alpha?

    b. Concern about the dynamics. In ALE v5, if I understand correctly, the dynamics are stochastic because of sticky actions. How does your algorithm break down with increasingly stochastic environments? (also, see questions below)

    c. Continuous actions: It appears the SAC style algorithm may work with continuous actions. Do you see any immediate hurdles to this? I don't see why MC samples are needed (cf SAC1,2)

If you can address these weaknesses and questions below, I am happy to raise my score.

[1]: https://arxiv.org/pdf/2109.11867

**Questions:**

This is a very interesting paper, so I am genuinely curious about the answers to the questions below. I think addressing some answers in the paper may also improve its impact.

0. Thermodynamic interpretation. For MaxEnt/KL RL, there is a natural interpretation of the trajectory distribution as the Boltzmann form, minimizing the free energy. Is there any similar interpretation here? I guess the more fundamental question is, what is the trajectory distribution, and what is the optimality variable that will yield the corresponding optimal distribution according to this measure of entropy?

1. Do you think the parameter $\alpha$ can be learned? You hint at it for future work, but what mechanism could be applicable? The same as used for $\beta$?

2. Conceptual: Use of alpha vs. beta. Can the authors explain a bit more about how the alpha parameter operates differently than beta? I.e. for a fixed alpha=1, is it possible that tuning beta gives you more or less the same result?

3. Stochasticity: Stochastic dynamics are an important feature in many RL environments. Do you have any ideas how to approach this problem in your framework? How might it connect to section 3 of [2]? Using the unconstrained solution method, [3] provides a solution for the constrained dynamics problem; can it work in your setting?

4. I'm having trouble seeing the difference between alpha-SQL and SQL. Can you explain if there's a difference?

5. Is there any connections to the Renyi divergence as used in compositionality [4]?

6. Do you really use FILO replay buffer? Why not the more common FIFO?

7. Does your baseline algorithm of SAC discrete also use $K$ critics? Does this framework suffer from overestimation bias? Also, does the baseline algo use a learned temperature parameter as well?

7. I'm very curious about Eq 29; do you face numerical instabilities? Esp wrt $e^{\beta Q}$ if $Q \sim (1-\gamma)^{-1}$ or $\pi <<1$?

8. L408 "set to be a multiple of $\log |A|$", what multiple? How did you decide?

9. L478, can you provide a reference for the original formulation ?

Very minor:
- I think I'm missing a step in my derivation of equations 13, 14; could you provide it in the appendix?
- $\gamma \in (0,1) \to \gamma \in [0,1)$
- inconsistent use of "entropy-regularised" vs "entropy regularised"
- Notation of base policy $\pi_b$ is usually called prior policy with $\pi_0$ especially wrt entropic measures
- more common to put factor of $\beta^{-1}$ on KL term, not $\beta$ on r.
- sometimes $\alpha$ is used as $\beta^{-1}$ e.g. in SAC [5]
- in sec 2.1 maybe mention "MaxEnt" as it is a familiar term
- L035 typo "a approximate" -> "an approximate"
- L047 typo "probablistic"
- L132 typo "by joint" -> "by the joint"
- L165 typo "policy policy"
- L327 typo "both make use a collection"
- L403 extra space before "."

[2]: https://arxiv.org/pdf/1805.00909
[3]: https://proceedings.mlr.press/v216/arriojas23a/arriojas23a.pdf
[4]: https://arxiv.org/pdf/1812.02216
[5]: https://arxiv.org/pdf/1801.01290

---

### Official Review · Reviewer_39J5 · 2024-10-31

**Soundness:** 2
**Presentation:** 4
**Contribution:** 1
**Rating:** 3
**Confidence:** 5

**Summary:**

Starting from the RL as inference framework, the authors investigate a space of algorithms derived minimising the $\alpha$-Renyi divergence to the underlying trajectory inference problem. This generalises the commonly used KL divergence. The authors derive the corresponding $\alpha$-Renyi Bellman operators and prove contraction properties. An algorithm is developed and evaluated on four discrete control tasks.

**Strengths:**

The paper is well-written and clearly differentiates itself from related work. The authors derive the algorithm clearly.

Theoretical Strengths

The authors provide a thorough theoretical analysis of the contraction properties of their derived Bellman operators. They characterise the corresponding optimal Boltzmann policy derived from their framework.

**Weaknesses:**

Empirical weakness:

My main concern is that the RL as inference/divergence regularised RL framework yields a huge family of algorithms including TRPO and it's simplified approximation, PPO. Out of this zoo of possible algorithms, PPO and REDQ have emerged as the go-to for researchers looking to solve control tasks where reward is dense. By introducing a new set of algorithms into this family, I'd expect the authors to compare to several approaches that attain strong empirical performance over a range of domains, including PPO and REDQ as a minimum.

Moreover, results are only shown in four Atari domains. I don't think this constitutes a significant number of domains to deduce improved performance of their proposed algorithm.

Algorithmic weakness:

The authors require that the action space is discrete. This renders their approaches only suitable for discrete control. The fact that their actor-critic derived approach cannot be applied to continuous control problems that other approaches naturally extend to is a major disadvantage.

**Questions:**

Can the authors show their approach offers a significant Pareto improvement in performance over a much larger range of domains when compared to several strong algorithms (a bare minimum of PPO and REDQ) that have been developed for control?

Can the authors extend their algorithms to continuous control problems,

 If convinced by responses to these questions, I'm happy to raise my score.

---

### Official Review · Reviewer_SM9K · 2024-11-05

**Soundness:** 3
**Presentation:** 2
**Contribution:** 2
**Rating:** 6
**Confidence:** 3

**Summary:**

This submission extends previous work on KL-regularized RL to Rényi divergences instead of the KL term. It derives the Rényi (regularised objective) via Rényi variational inference and discusses the relation between the regularisation and the inference views, introduces theoretical results for PE and PI and demonstrates two practical algorithms, $\alpha$-SAC and $\alpha$-SQL.

**Strengths:**

- good example of an extension to the KL-regularized RL framework
- well structured, clearly written, easy to follow, I particularly liked the way this work was contextualized within the regularization/inference views

**Weaknesses:**

- the _practical_ policy evaluation algorithms derived in this paper only hold for the case with deterministic dynamics (although the Bellman operator is general). Not a problem in itself necessarily but maybe the two algorithms should have been evaluated on deterministic environments and not Atari v5 which has sticky actions for inducing stochastic dynamics.
- relatedly, it would have been interesting to characterize or discuss more what is lost with this assumption
- although I liked the way the paper is organized and how the new divergence is introduced into the broader framework, I believe this work could benefit from a wider discussion in relation with past works, eg. [1]
- the empirical results seem a bit underwhelming, especially in the absence of a (temperature) tuned SQL baseline

[1] https://arxiv.org/abs/2003.14089

**Questions:**

- How were the 4 games selected? There's work suggesting we can make a principled selection of 3,5,10 atari games that would make a good predictor of the overall performance [1]
- Why not include soft q-learning in the comparison?

[1] https://arxiv.org/abs/2210.02019

---

### Note · Authors · 2024-11-26

I have read and agree with the venue's withdrawal policy on behalf of myself and my co-authors.